# The *FLNC* Ala1186Val Variant Linked to Cytoplasmic Body Myopathy and Cardiomyopathy Causes Protein Instability

**DOI:** 10.3390/biomedicines12020322

**Published:** 2024-01-30

**Authors:** Marion Onnée, Audrey Bénézit, Sultan Bastu, Aleksandra Nadaj-Pakleza, Béatrice Lannes, Flavie Ader, Corinne Thèze, Pascal Cintas, Claude Cances, Robert-Yves Carlier, Corinne Metay, Mireille Cossée, Edoardo Malfatti

**Affiliations:** 1Institut Mondor de Recherche Biomédicale, Université Paris Est Créteil, Institut National de la Santé et de la Recherche Médicale U955, 94010 Créteil, France; marion.onnee@inserm.fr (M.O.); sultan.bastu@inserm.fr (S.B.); 2Neurologie et Réanimation Pédiatrique, Assistance Publique–Hôpitaux de Paris, Université Paris Saclay, Département Médico-Universitaire Santé de l’Enfant et de l’Adolescent, Hôpital Raymond Poincaré, 92380 Garches, France; audrey.benezit@aphp.fr; 3Centre de Référence des Maladies Neuromusculaires Nord Est Ile-de-France, Service de Neurologie, Hôpitaux Universitaires de Strasbourg, 67200 Strasbourg, France; aleksandra.nadaj-pakleza@chru-strasbourg.fr; 4European Reference Network, EURO-NMD, Neuromuscular Centre at Hautepierre Hospital, Hôpitaux Universitaires de Strasbourg, 67200 Strasbourg, France; 5Département de Pathologie, Hôpitaux Universitaires de Strasbourg, 67091 Strasbourg, France; beatrice.lannes@chru-strasbourg.fr; 6Assistance Publique–Hôpitaux de Paris, Sorbonne Université, Département Médico-Universitaire BioGem, Unité Fonctionnelle de Cardiogénétique et Myogénétique Moléculaire et Cellulaire, 75013 Paris, France; flavie.ader@aphp.fr; 7Institut National de la Santé et de la Recherche Médicale UMRS1166, Université Paris Cité, 75006 Paris, France; 8Laboratoire de Génétique Moléculaire, Centre Hospitalier Universitaire de Montpellier, Université de Montpellier, 34095 Montpellier, France; corinne.theze@inserm.fr; 9Centre de Référence des Maladies Neuromusculaires AOC (Atlantique-Occitanie-Caraïbes), Département de Neurologie, Hôpital Pierre-Paul Riquet, Centre Hospitalier Universitaire de Toulouse, 31059 Toulouse, France; cintas.p@chu-toulouse.fr (P.C.); m-cossee@chu-montpellier.fr (M.C.); 10Centre de Référence des Maladies Neuromusculaires AOC (Atlantique-Occitanie-Caraïbes), Unité de Neurologie Pédiatrique, Hôpital des Enfants, Centre Hospitalier Universitaire de Toulouse, 31059 Toulouse, France; cances.c@chu-toulouse.fr; 11Assistance Publique–Hôpitaux de Paris, Groupe Hospitalier Universitaire Paris Saclay, Département Médico-Universitaire Smart Imaging, Service d’Imagerie Médicale, Institut National de la Santé et de la Recherche Médicale UMR1179, Hôpital Raymond Poincaré, 92380 Garches, France; robert.carlier@aphp.fr; 12Unité Fonctionnelle de Cardiogénétique et Myogénétique Moléculaire et Cellulaire, Centre de Génétique Moléculaire et Chromosomique, Sorbonne Université, Institut National de la Santé et de la Recherche Médicale, Institut de Myologie, Groupe Hospitalier La Pitié-Salpêtrière, 75013 Paris, France; corinne.metay@aphp.fr; 13PhyMedExp, Université de Montpellier, Institut National de la Santé et de la Recherche Médicale, Centre National de la Recherche Scientifique, 34295 Montpellier, France; 14Assistance Publique–Hôpitaux de Paris, Centre de Référence de Pathologie Neuromusculaire Nord-Est-Ile-de-France, Hôpital Henri Mondor, 94000 Créteil, France

**Keywords:** cardiomyopathy, myopathy, *FLNC*, mutation

## Abstract

Filamin C-related disorders include myopathies and cardiomyopathies linked to variants in the *FLNC* gene. Filamin C belongs to a family of actin-binding proteins involved in sarcomere stability. This study investigates the pathogenic impact of the *FLNC* c.3557C > T (p.Ala1186Val) pathogenic variant associated with an early-onset cytoplasmic body myopathy and cardiomyopathy in three unrelated patients. We performed clinical imaging and myopathologic and genetic characterization of three patients with an early-onset myopathy and cardiomyopathy. Bioinformatics analysis, variant interpretation, and protein structure analysis were performed to validate and assess the effects of the filamin C variant. All patients presented with a homogeneous clinical phenotype marked by a severe contractural myopathy, leading to loss of gait. There was prominent respiratory involvement and restrictive or hypertrophic cardiomyopathies. The Ala1186Val variant is located in the interstrand loop involved in intradomain stabilization and/or interdomain interactions with neighbor Ig-like domains. 3D modeling highlights local structural changes involving nearby residues and probably impacts the protein stability, causing protein aggregation in the form of cytoplasmic bodies. Myopathologic studies have disclosed the prominent aggregation and upregulation of the aggrephagy-associated proteins LC3B and p62. As a whole, the Ala1186Val variant in the *FLNC* gene provokes a severe myopathy with contractures, respiratory involvement, and cardiomyopathy due to protein aggregation in patients’ muscles.

## 1. Introduction

The *FLNC* gene encodes the filamin C (*FLNC*) protein belonging to a family of actin-binding proteins involved in sarcomere stability maintenance. *FLNC* is predominantly expressed in both skeletal and cardiac muscles and localized to Z-discs, myotendinous junctions, sarcolemma, and intercalated discs [1,2,3]. *FLNC* consists of 24 Ig-like repeats, divided into two subdomains (ROD1 and ROD2), and presents two intervening calpain-sensitive hinges necessary for *FLNC* subunit mobility and dimerization regulation [4]. The protein has a N-terminal domain that cross-links actin filaments (F-actin) and a C-terminal part that interacts with binding partners and allows for the dimerization of the protein (by the last Ig repeat, Ig24), essential for its function. *FLNC* has numerous binding partners and is involved in several biological functions, such as the organization and stability of sarcomeres and Z-disc assembly, and it has a role in transmitting/responding to the mechanical forces acting on the F-actin network. For more detailed information about the role and structure of *FLNC*, see the reviews in [4,5,6].

*FLNC* variants were first associated with myopathies with excessive intracellular protein aggregate accumulation, the myofibrillar myopathies (MFM) in 2005 [7]. Successively, *FLNC* variants have been identified in patients with distal myopathies (DM) [8,9], more often associated with haploinsufficiency, and very recently, in patients with congenital myopathies (CM) [10]. More recently, *FLNC* variants have also been associated with isolated cardiomyopathies [11,12,13]. To date, *FLNC* variants have been associated with different disorders, ranging from skeletal muscle myopathies (MFM, DM or CM) to cardiomyopathies (MIM# 102565) [4]. *FLNC* variants are mostly reported in the adult population, with little data available in pediatric cohorts. Notably, only a few cases have been reported with a concomitant manifestation of cardiac and skeletal muscle pathologies.

Here, we thoroughly describe three patients with the *FLNC* (NM_001458.5) c.3557C > T (p.Ala1186Val) variant associated with early-onset myopathy with contractures, restrictive or hypertrophic cardiomyopathy, and cytoplasmic bodies in muscle biopsies. Functional validation studies showed that *FLNC* is found in protein aggregates due to altered *FLNC* stability linked to the *FLNC* Ala1186Val pathogenic variant.

## 2. Materials and Methods

### 2.1. Patients

This study included three unrelated patients presenting with a similar phenotype referred to the Neuromuscular Pathology Reference Centers of Henri Mondor Hospital, Raymond-Poincaré Hospital, Toulouse University Hospital, and the University Hospital of Strasbourg. This study was conducted in accordance with the Declaration of Helsinki. Informed consent was obtained from all subjects involved in this study.

### 2.2. Targeted Gene Enrichment and Next-Generation Sequencing (NGS)

Each patient’s DNA was extracted from peripheral blood with QIAsymphony (Qiagen, Hilden, Germany) and qualitatively checked using the Tape Station DNA genomic array (Agilent, Santa Clara, CA, USA). Custom targeted gene enrichment and DNA library preparation were performed using the Nimblegen EZ choice probes and Kappa HTP Library preparation kit according to the manufacturer’s instructions (Nimblegen, Roche Diagnostics, Madison, WI, USA). A specific custom panel of 17 genes was designed, including genes associated with myofibrillar myopathies. The RefSeq coding sequences were determined to be consensual for genetic diagnosis within a French nationwide working group [14]. The targeted regions included all coding exons and ± 50 base pairs of the flanking intronic regions of 16 genes known to be involved in myofibrillar myopathies (*ACTA1* (NM_001100.3), *BAG3* (NM_004281.3), *CRYAB* (NM_001885.2), *DES* (NM_001927.3), *DNAJB6* (NM_058246.3), *FHL1* (NM_001159702.2), *FLNC* (NM_001458.4), *GNE* (NM_001128227.3), *HSPB1* (NM_001540.3), *HSPB8* (NM_014365.2), *MYH2* (NM_017534.5), *MYOT* (NM_006790.2), *SQSTM1* (NM_003900.4), *TTN* (NM_001267550.1), *VCP* (NM_007126.3), and *ZASP/LDB3* (NM_001080114.1, NM_007078.2,NM_001171610.1). Paired-end sequencing was performed on a 250-cycle Flow Cell (Illumina, Santa Cruz, CA, USA) using the Illumina MiSeq platform. Eight libraries were multiplexed per run.

### 2.3. Bioinformatics Analysis

MiSeq Software v.4.0 (Illumina, San Diego, CA, USA) generated the FASTQ format files after demultiplexing patients’ sequences. Sequence alignment against the human reference genome (Hg19) was performed using BWA-MEM. Variant calling was performed using the GATK Haplotype Caller program. Detected variants were then annotated using the ANNOVAR and CADD tools. Detected variants with sequencing depths greater than 30X and for which at least 20% of reads supported the alternative allele were kept for analysis. The detection of copy number variation (CNV) was performed after coverage normalization by computing the ratio of a target’s coverage of a given individual over the mean coverage of this target across all patients in the same sequencing run.

### 2.4. The Variant’s Interpretations

The pathogenicity of variants was determined according to the current American College of Medical Genetics and Genomics (ACMG) guidelines [15]. Variants were filtered out according to their allele frequency (≤1%), as reported in the Genome Aggregation Database (gnomAD) database (http://gnomad.broadinstitute.org/). We then evaluated each variant by considering a review of the literature, the location of the variant in the gene, and the resulting corresponding protein. In addition, we looked at a local database of pathogenic variants related to our experience regarding the molecular diagnosis of myopathies. All variants considered to be pathogenic and likely pathogenic were confirmed via a second independent method (Sanger sequencing).

### 2.5. Protein Structure Analysis

The alignment of the immunoglobulin-like repeat 10 (Ig10) of all 3 human filamins, (A (NP_001447.2), B (NP_001157789.1), and C (NP_001449.3)), were rendered using ESPript3 [16]. The sequence of human *FLNC*-Ig10 was searched against known 3D structures using PHYRE2 [17]. Alignments with human FLNA-Ig10 (PDB 3RGH) [18] was proposed with the highest scores (considering only crystal structures), covering the entire sequence of the query (confidence 99.9%, 63% sequence identity). SWISS-MODEL [19] was used to model the human *FLNC*-Ig10 to depict amino acid position 1186, using the 3RGH 3D structure as a template. This model was manipulated using Chimera [20].

### 2.6. Muscle Morphological and Protein Studies

#### 2.6.1. Histochemistry and Immunohistochemistry

Conventional histochemical techniques were performed for hematoxylin and eosin (HE), Engel–Gömöri trichrome, cytochrome C oxidase (COX), nicotinamide adenine dinucleotide (NADH), menadione-linked αglycerophosphate dehydrogenase, and the immunostaining of fast myosin heavy chain (MHC-fast), ubiquitin, desmin, and dystophin. Digital photographs of each biopsy were obtained using a Zeiss AxioCam HRc linked to a Zeiss Axioplan Bright Field Microscope and processed using Axio Vision 4.4 software (Zeiss, Jena, Germany).

#### 2.6.2. Immunofluorescence

Next, 7 µm of thick cryostat sections was collected on Super Frost Plus slides (Thermo Fisher Scientific, Waltham, MA, USA), permeabilized for 5 min with Triton 0.5% and blocked in 10% BSA for 1 h at room temperature. Then, the slides were incubated overnight at 4 °C with the following primary antibodies: anti-*FLNC* (rabbit, NBP1-89300, Novus Biologicals (Centennial, CO, USA), 1:100), anti-p62 (mouse IgG1, 610832, BD biosciences (Franklin Lakes, NJ, USA), 1:100), and anti-LC3B (mouse IgG2a, ab243506, Abcam (Cambridge, UK), 1:200). The next day, after repetitive washes, the slides were incubated with Alexa fluor secondary antibodies (anti-rabbit IgG-Alexa 488 (A11034), anti-mouse IgG1-Alexa 555 (A12227), and anti-mouse IgG2a-Alexa 647 (A21241), all sourced from Thermo Fisher Scientific (Waltham, MA, USA), 1:300) for 1 h at room temperature. Laminin staining was performed after the secondary antibody incubation for 2 h at 37 °C using a conjugated antibody (NB300-144AF647, Novus Biologicals, Centennial, CO, USA). The slides were finally incubated with DAPI (1 µg/mL) for 5 min at room temperature. Histologically normal muscles were used as controls. Images were obtained using a Zeiss AxioCam Mrm linked to a Zeiss AxioImager D1 Microscope and processed using Zen Blue 2.0 software (Zeiss, Jena, Germany). Image analyses were performed using Fiji software v.2.9.0 (https://fiji.sc/). The JACoP (https://imagej.net/plugins/jacop) and Colocalization Finder plugins were used to analyze the colocalization and to calculate the Pearson coefficient. We measured the Pearson coefficient between *FLNC* and LC3B or p62 staining after the manual isolation of cytoplasmic body-like structures from about 400 fibers.

#### 2.6.3. Electron Microscopy

Muscle samples were washed in 0.15 M cacodylate buffer and postfixed for 2 h in 2% osmium tetraoxide in 0.1 M cacodylate buffer. After washing in cacodylate buffer and water, samples were dehydrated in increasing concentrations of ethanol, with the final dehydration occurring in 100% acetone and using propylene oxide. To avoid humidity, the samples were infiltrated with 1/3 Epon and 2/3 propylene oxide. Finally, they were infiltrated 3 times with pure Epon and then placed into molds with fresh Epon at 60 °C for 24 h. Semi-thin (0.5 µm thick) and ultrathin (70 nm thick) transverse sections were cut with a Leica UC7 ultramicrotome (Leica, Leica Microsystemes SAS, Nanterre, France). The semi-thin sections were stained with 1% toluidine blue in 1% borax, and the region of interest was selected under the microscope. The ultrathin section of the selected region was collected using copper grids (200 mesh, EMS, Souffelweyersheim, France) and contrasted with Reynold’s lead citrate. The ultrathin sections were observed using a Hitachi HT7700 electron microscope (Milexia, Saint Aubin, France) operating at 100 kV. Pictures (2048 × 2048 pixels) were taken with an AMT41B camera.

#### 2.6.4. Western Blot

Next, 50× 10 µm thick cryosections of muscle tissues were placed in a Lysing Matrix D 2 mL tube (#6913500, MP Biomedicals, Santa Ana, CA, USA) containing 300 µL of radioimmunoprecipitation assay buffer (RIPA) lysis buffer (50 mM Tris-HCl pH8, 150 mM NaCl, 1 mM EDTA pH8, 1% NP40, 0.5% sodium deoxycholate, 0.1% SDS). The samples were homogenized using a ribolyzer. The homogenate was then centrifuged at 20,000× *g* for 10 min at 4 °C. For RIPA-soluble proteins, the supernatant was then isolated, and the protein concentration was determined using the BCA protein assay kit (Cat# 23227, Pierce, Appleton, WI, USA). For RIPA-insoluble proteins, the pellet was resuspended in 100 µL of urea lysis buffer (8 M urea, 2 M thiourea, 3% SDS, 0.05 M Tris-HCl, 0.03% bromophenol blue, pH 6.8). The homogenate was then centrifuged at 20,000× *g* for 10 min at 4 °C. The supernatant was then isolated, and the protein concentration was determined using the Pierce™ 660 nm protein assay kit (Cat# 22660, Pierce, Appleton, WI, USA). Then, for RIPA-soluble and -insoluble proteins, 10 µg of protein was denatured for 10 min at 70 °C and separated on a 4–12% SDS-PAGE gel (Cat# 345-0009, Bio-Rad, Hercules, CA, USA) and subsequently transferred to polyvinylidene difluoride (PVDF) membranes. We colored the membrane with Ponceau Red to check for the presence of proteins. The membrane was blocked with 5% non-fat dry milk in phosphate-buffered saline (PBS) containing 0.05% Tween20 (PBS-T) for 1 h at room temperature, before being incubated overnight at 4 °C with primary antibodies (anti-*FLNC*, NBP1-89300, Novus Biologicals (Centennial, CO, USA), 1:500 and anti-beta actin, ab184220, abcam (Cambridge, UK), 1:1000). After washes with PBS-T, the membrane was incubated for 1 h at room temperature with secondary antibodies (anti-mouse IgG-HRP, sc-2055, 1:2500 and anti-rabbit IgG-HRP, sc-2054, 1:10,000, both from Santa Cruz Biotechnology, Dallas, TX, USA). Detection was performed using the ECL Plus Western Blotting Detection System (Cat# PRN2132, GE Healthcare, Chicago, IL, USA). Protein band densities were determined using Fiji software v2.9.0 (https://fiji.sc/). Beta actin served as the loading control.

## 3. Results

### 3.1. Patients’ Characteristics

Patient 1 (P1) is a 17-year-old male born at term to healthy consanguineous parents, following a normal uncomplicated pregnancy. His sister presents congenital adrenal hyperplasia. P1 showed multiple examples of arthrogryposis affecting the shoulders, elbows, fingers, hips, and knees, as well as high-arched palate and microretrognathia. The neonatal period was unremarkable, and cardiac workup was normal at 2 and 3 years old. P1 began to show learning difficulties at school, and at around 10–11 years, he developed progressive scoliosis and restrictive respiratory insufficiency. At this time, the forced vital capacity (FVC) was 800 mL. Serum creatine kinase (CK) levels were normal, as was electromyography (EMG). The patient developed progressive amyotrophy, particularly in the scapular muscles, as well as chewing difficulties, limited mouth opening, and limited arytenoid mobility. At 12 years, he showed mild global muscular weakness with elbow flexion contractures of 10–15°, a very rigid kyphosis, a small thorax, and pectus carinatum. At the same age, he underwent posterior spinal fusion, which was followed by a severe postoperative course, including hemorrhagic shock complicated by acute respiratory distress syndrome and septic shock needing extracorporeal membrane oxygenation. P1 required tracheostomy and gastrostomy. He had a progressive course with major diffuse muscular contractures, the loss of the ability to walk, and weakness in the upper limbs, as well increasing amyotrophy. Cardiac workup at 13 years old revealed a restrictive cardiomyopathy (RCM). EMG was myopathic at this time. The comparison before and after surgery to treat the spinal deformities is illustrated in Figure 1. Whole-body muscle magnetic resonance imaging (MRI) showed, on water images of the T2 Dixon sequence, massive hyperintensity affecting the muscular bodies of the scapular girdle, pectoral muscles, gluteal muscles, quadriceps, hamstring muscles, and short and long adductor muscles, as well as pre-existing diffuse muscle atrophy, consistent with a neuro-myopathy of reanimation on diffuse muscle atrophy associated with arthrogryposis sequelae (Figure 2). The last examination performed in 2022 showed a significant aggravation of overall muscle atrophy, predominantly affecting the erector muscles of the dorsal and lumbar spine, the psoas muscles, and, to a lesser extent, the pelvic floor and gluteal muscles (Figure 3).

Patient 2 is a 43-year-old man born to healthy unrelated parents. He presented contractures in the Achilles tendon from the age of 2 years old. He successively developed progressive muscular weakness with muscle contractures, and he lost gait at 30 years of age. He had severe scoliosis treated with spine fusion, and he developed severe pulmonary restrictive syndrome, requiring a tracheostomy at 36 years of age. Hypertrophic cardiomyopathy (HCM) was detected at the age of 39 years old. His CK levels were 1000 UI/L.

Patient 3 is a 34-year-old woman born to healthy unrelated parents. She presented toe-walking at 3–4 years old; she developed multiple contractures with rigid spine, scoliosis, and mild axial and proximal motor weakness. At her last examination at 34 years old, she could walk 200 m, and her CK levels were 4–7 times higher than normal values. P3 developed a restrictive syndrome with FVC at 1.33 L, 38% of the normal value, thus requiring noninvasive nocturnal ventilation. At 29 years old, she developed HCM with relapsing atrial fibrillation, and she underwent preventive pace-maker implantation at 33 years of age.

### 3.2. Muscle Biopsy Findings

P1 underwent a vastus lateralis muscle biopsy at 12 years of age that showed nuclear internalizations, rimmed vacuoles, and the presence of multiple variably sized cytoplasmic bodies, sometimes associated with rare rimmed vacuoles and amorphous filamentous inclusions (Figure 4A,B). The protein inclusions misplaced the mitochondrial network (Figure 4C). We also observed pronounced fiber size variation with highly atrophic type 2 fibers and hypertrophy of type 1 fibers, as well as the predominance of type 1 fibers (Figure 4D). A P2 muscle biopsy showed the presence of multiple cytoplasmic bodies. Unfortunately, there was no muscle available for further studies. P3 deltoid muscle biopsy showed a similar myopathologic picture, including internalized nuclei and cytoplasmic bodies (Figure 4E,F). Interestingly, fibers presenting cytoplasmic bodies harbored areas containing inclusions with increased immunoreactivity for desmin (Figure 4G). The hypertrophy of type 1 fibers, atrophy of type 2 fibers, and predominance of type 1 fibers are also observed (Figure 4H).

Electron microscopy on P1 muscle biopsy showed multiple cytoplasmic bodies, consisting of central dense osmiophilic areas, with a radiating filamentous material halo (Figure 5A,B). There were also massive sarcomeric disruptions, abnormally clustered mitochondria, and lipofuscin residual bodies (Figure 5C,D).

### 3.3. Genetic Testing Results

Targeted NGS analyses revealed the missense heterozygous variant in the *FLNC* gene (NM_001458.5), c.3557C > T (p.Ala1186Val), located in exon 21, more specifically in immunoglobulin-like domain 10 (Figure 6A). The Ala1186 residue is highly conserved in 12 species and located in a highly conserved region of the protein, corresponding to the immunoglobulin-like domain (Figure 6B). This variant is reported in the ClinVar (Variation ID: 427928) and dbSNP (rs1114167361) databases but is absent from gnomAD.

### 3.4. Impact of the FLNC Variant in the Protein Tertiary Structure

The residue Ala1186 is localized in the conserved immunoglobulin-like repeat 10 (R10) domain of *FLNC* (*FLNC*-Ig10) (Figure 6A), which remains unsolved using cryo-electron microscopy or crystallography. Since the *FLNC*-Ig10 domain shares more than 60% of its identity with FLNA- and FLNB-Ig10 (Figure 6C), we considered the crystal structure of the human FLNA-Ig10 protein (PDB 3RGH, 2.44 Å resolution) when building a model of the human *FLNC*-Ig10 protein using SWISS-MODEL (Figure 6D). The mutated amino acid Ala1186 is in a loop between two ꞵ-strands (loop BC), indicated in purple in the 3D model. The Ala1186Val variant involves a residue located between the two beta sheets, and it plays a role in their stability thanks to hydrophobic C-C interactions with two nearby non-polar amino acid residues, Gly1231 and Phe1153 (Figure 6E). In addition to carbon–carbon interactions, each hydrogen atom from CH3 side chain of Ala1186 residue forms at least two Van der Waals hydrophobic interactions with Phe1153 and Tyr1230 aromatic residues. The first one is in the A strand in the first beta sheet, and the second one is in the F strand in the second beta sheet. The hydrophobic zone formed by these three residues may play a role in interactions with neighboring Ig-like domains, even if we still do not know how Ig-like domains interact with each other within the whole *FLNC* protein. Indeed, hydrophobic interaction is the main driving force behind protein folding, and it critically affects stability and solubility. Then, the Ala1186 residue forms two hydrogen bonds with Tyr1230, stabilizing the two beta sheets together. Even if the physicochemical difference is small between arginine and valine, with both being non-polar, the alanine-to-valine substitution, with valine being bigger, could affect the structure/stability of the hydrophobic zone. Steric clashes of valine insertion cause local structural changes involving the three nearby residues (Figure 6F). After the minor minimization of these residues, we notice a small increase in the distance between the atoms forming hydrogen bonds and greatly increased hydrophobic contacts with aromatic residues (Figure 6G).

### 3.5. FLNC Protein Expression in Muscle Biopsies

Immunostaining using anti-*FLNC* antibody revealed strong positivity for *FLNC* in areas corresponding to protein aggregates and sarcoplasmic accumulations (Figure 7A). Immunoblotting studies did not reveal any differences in the expression of soluble fractions of *FLNC* in patients compared to the controls (Figure 7B). However, we observed accumulations of the insoluble fraction of *FLNC* protein in patients compared to controls, consistent with the formation of protein aggregates (Figure 7C). We showed, in each patient’s muscle sections, an abnormal accumulation of p62 and LC3B proteins in structures with a strong *FLNC* signal, which can correspond to cytoplasmic bodies (Figure 8A). *FLNC* proteins mainly accumulate to the edges of cytoplasmic bodies, highly colocalizing with LC3B proteins (Figure 8B), whereas p62 is mainly localized inside cytoplasmic bodies containing *FLNC* and other protein aggregates (Figure 8C).

## 4. Discussion

Extensive genetic research into cardiac and neuromuscular disorders has revealed the *FLNC* gene’s significant role in various pathologies affecting both striated skeletal muscle cells and cardiomyocytes due to its multiple biological functions. *FLNC* variants were first associated with neuromuscular diseases such as MFM or DM, and they have since been associated with cardiomyopathies. Only a few past studies reported a concomitant manifestation of cardiac and skeletal muscle pathologies associated with *FLNC* variants [10,21]. The simultaneous occurrence of cardiac and skeletal muscle manifestations is not surprising since the *FLNC* gene is expressed in both tissues.

In our cohort, Patient 1 presented with arthrogryposis at birth, and all patients developed a highly contractural phenotype with progressive muscular weakness, leading to respiratory involvement and gait loss. The cardiac involvement appeared later and was characterized by impaired ventricular filling due to a stiffening of the heart muscle, defined as restrictive cardiomyopathy, or an abnormal thickening of the heart’s walls, defined as hypertrophic cardiomyopathy, both of which lead to reduced cardiac function. Interestingly, the Ala1186Val variant has been identified in 13 patients reported in the literature, and 10 of them have a combination of myopathy with cardiomyopathy [10,21,22,23,24,25,26,27]. As a whole, the disease trajectory of filaminopathy associated with the Ala1186Val variant consisted of a contractural phenotype at birth or during the first year of life, followed by the development of cardiomyopathy, usually diagnosed during the first three years of life [10,21,22,23,26]. The trajectory described in our three patients is consistent with the other previously reported cases, whose characteristics are shown in Table 1. In particular, our cases show different ages of onset with later-onset cardiomyopathies, both for restrictive and hypertrophic cases. Thus, we can define the amino acid substitution Ala1186Val as a mutational hotspot for the skeletal muscle myopathy and cardiomyopathy combination, as suggested by Muravyev and colleagues [21]. As *FLNC* is expressed at the myotendinous junction [2], we could imagine an increased stiffness due to its altered conformation linked to the Ala1186Val variant. We could not observe myotendinous junctions in patients’ muscle biopsies. Further studies are necessary to validate this hypothesis.

Muscle MRI in P1 demonstrates distinctive changes, including symmetrical lipomatous alterations in various thigh and lower leg muscles. Notably, certain muscles show relative preservation, while others exhibit extensive fatty degeneration. Here, we show the evolution of muscle MRI features of P1 consisting of hyperintensity in T2 Dixon water images, including the scapular girdle, pectoral, gluteal, quadriceps, hamstrings, and adductor muscles. This condition is linked to neuro-myopathy, possibly due to intensive care, in the context of existing diffuse muscle atrophy and arthrogryposis sequelae. A 2022 follow-up study indicated worsening muscle atrophy, mainly affecting the dorsal and lumbar erector muscles, psoas muscles, and, to a lesser extent, the pelvic floor and gluteal muscles. The consistent observation of these distinct muscle involvement patterns across multiple studies underscores the reliability of MRI as a diagnostic tool for *FLNC*-related myopathies [8,28,29,30,31,32].

Among the 13 patients carrying the Ala1186Val variant described in the literature, images of histopathological findings of routine stains and electron microscopy are available for only one patient [24]. The myopathologic findings of our patients shared significant similarities to those previously described by Matsumura and colleagues. Indeed, we identified cytoplasmic bodies, myofibrillar abnormalities with amorphous inclusions, desmin accumulation in some myofibers, Z-disk disorganization, and abnormally clustered mitochondria. Muscle biopsies also presented major fiber size diameter variability, with type 1 fibers’ predominance and hypertrophy and type 2 fibers’ atrophy associated with internalized nuclei. *FLNC* immunohistochemistry in P1 revealed massive protein aggregation in myofibers consistent with the presence of cytoplasmic bodies and disruption of the sarcomeric structure observed during ultrastructural studies. Cytoplasmic bodies are protein inclusion bodies containing Z-disk material and striated muscle-specific intermediate filament proteins [33,34,35,36]. We also evidenced in P3 an accumulation of desmin (DES) within or close to cytoplasmic bodies. Interestingly, mutations in *DES* often lead to the formation of intracellular aggregates, disrupting the normal architecture of muscle cells [13,37,38]. As discussed by Brodehl and colleagues, this aberrant aggregation is linked to a variety of myopathies and cardiomyopathies, similar to what is observed with *FLNC* mutations [13]. Alpha B Crystallin, encoded by the *CRYAB* gene, is a small heat shock protein that plays a crucial role in protein folding and aggregation inhibition. Mutations in *CRYAB* can disrupt its chaperone function, leading to protein misfolding and aggregation, which is a common pathologic feature in certain myopathies and cardiomyopathies [39,40]. Like *FLNC* and *DES*, the *CRYAB* mutations underscore the importance of protein stability and proper folding for maintaining cardiac muscle function. The comparison of *FLNC* with *DES* and *CRYAB* highlights a shared pathological mechanism in cardiomyopathies: the disruption of normal protein folding and aggregation processes. While each gene and its corresponding protein have unique roles and structures, the overarching theme of their contribution to cardiomyopathy lies in their failure to maintain cellular integrity due to aberrant protein aggregation. This comparison not only broadens our understanding of the molecular basis of cardiomyopathies but also suggests potential therapeutic targets centered around protein stability and aggregation control in these three conditions.

Of note, in our study, the *FLNC* aggregates were found at the edges of cytoplasmic bodies forming a scaffold surrounding other protein aggregates. In order to study the composition of cytoplasmic bodies, we performed co-immunostaining of *FLNC* and the aggrephagy-associated proteins LC3B and p62. LC3B is a marker for autophagosomes, which are double-membraned structures involved in autophagy, a cellular process responsible for degrading and recycling cellular components [41]. The colocalization between *FLNC* aggregates and LC3B suggests that the *FLNC* aggregates are being targeted for autophagic degradation, but an impairment of the autophagy process can involve the accumulation of LC3B-positive phagosomes that possibly cannot fuse to lysosomes and, thus, remain longer in the sarcoplasm. p62, also known as SQSTM1, is a protein that typically accumulates when autophagy is impaired [41]. Its levels increase when autophagy cannot effectively clear aggregated or damaged proteins. Thus, the upregulation of the aggrephagy-associated proteins LC3B and p62 and their localization in *FLNC*-positive structures shed light on a role of aggrephagy in the physiopathology of the patients, which is largely unknown in cytoplasmic bodies’ myopathy. However, *FLNC* protein aggregates have been widely described in MFM pathology [7,9,24,32]. Of note, the presence of *FLNC* aggregates in some myofibrils and the disruption of the Z-discs’ striated pattern were described in zebrafish embryos carrying the *FLNC* Ala1186Val variant [10]. Therapies accelerating autophagy might be proposed in the models as potential therapeutic approaches.

In order to understand the origin of the *FLNC* protein aggregation observed in the patients’ muscle biopsies, we studied the impact of the Ala1186Val pathogenic variant on the *FLNC* protein structure, and little is known about its 3D conformation and the roles of the different subdomains. The C terminal domain is the most studied part of the protein since it involves Ig23 (PDB 2NQC) [42] and Ig24 (PDB 1V05) [43] repeats, essential for protein dimerization and function. Then, within the ROD1 subdomain (R1 to R15), only Ig4-5 (PDB 3V8O) [44] and Ig14-15 (PDB 7OUU) repeats have been studied via X-ray diffraction, and they have shown a specific compact conformation [45]. We still do not know how other Ig-like domains in the ROD1 are organized within the *FLNC* protein, including the Ig10 domain where the Ala1186Val variant is located. Indeed, the *FLNC*-Ig10 protein domain remains unsolved through cryo-electron microscopy or crystallography. In 2018, Kiselev and colleagues used the solution structure of the hFlnC-Ig14 (PDB 2D7M, solved by Nuclear Magnetic Resonance), sharing 48% identity with hFLNC-Ig10 domain, as a template to build a model of the *FLNC*-Ig10 protein domain [10]. Later, Xiao and Muravyev’s teams used as a template the crystal structure of the hFlnA-Ig3-5 (PDB 4M9P, 1.72 Å resolution), sharing 37% identity with hFLNC-Ig10 domain [21,26]. We found that the hFLNC-Ig10 domain shares 63% sequence identity with the 3D structure of the hFLNA-Ig10 protein (PDB 3RGH), which was used, in this study, to build a model of the *FLNC*-Ig10 protein. Using this model, we were able to provide insights into the impact of the mutation of the residue Ala1186 to valine on its interactions with surrounding amino acids. The variant is located in an interstrand loop likely involved in interdomain interactions with neighbor Ig-like domains and/or intradomain stabilization through connections between the two beta sheets formed by strands A, B, E, and D and C, F, and G. A mutation of such a residue involving many hydrophobic interactions probably alters beta sheets’ stability and interactions with neighbor Ig-like domains. The resulting local structural changes can potentially disrupt the protein stability and folding, which can partially explain *FLNC* aggregation in muscle fibers. Our results are consistent with another study that made a link between other variants located in the interstrand loops and cytoplasmic aggregates of *FLNC* in cardiac tissues in the context of RCM [13]. These findings highlight the importance of these loops for protein stability/folding, whether in cardiac or skeletal muscle.

## 5. Conclusions

In conclusion, this study of *FLNC* Ala1186Val-associated filaminopathy has shed light on the dual impacts of this variant on both the skeletal muscles and cardiac function. The disease trajectory typically begins with contractures in early life, followed by the development of cardiomyopathy. Muscle biopsy findings have shown cytoplasmic bodies, sarcomeric disruption, and the massive protein aggregation of *FLNC*- and aggrephagy-associated proteins in muscle fibers. The Ala1186Val variant disrupts critical hydrophobic interactions, potentially influencing protein stability and folding, as well as inter- and intradomain interactions. While the exact consequences of these changes are not fully understood, it is essential to investigate the impact of this variant on *FLNC* protein function. Notably, the FLNA-R10 domain has shown F-actin-binding activity [46] and shares high similarities with the *FLNC*-R10 domain. Further studies are needed to determine how a variant in this domain might affect *FLNC*’s function, including its potential F-actin-binding activity.

## Figures and Tables

**Figure 1 biomedicines-12-00322-f001:**
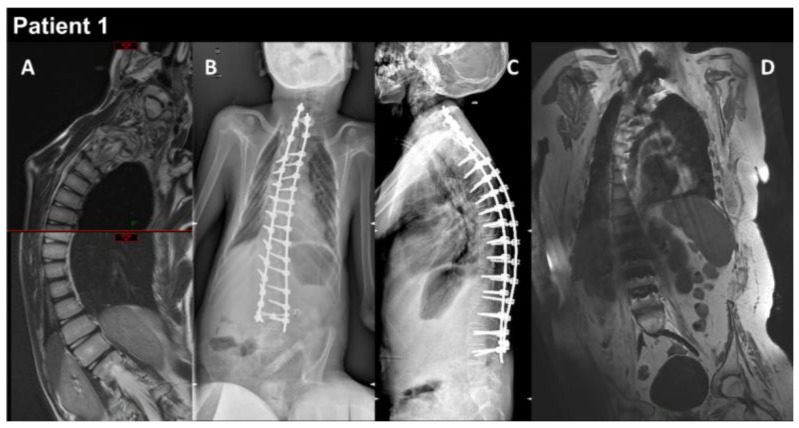
P1 MRI coronal views of the thoraco-lumbar spine before (**A**) and after (**B**–**D**) spine arthrodesis. Coronal and lateral radiographs of the spine in the patient in the sitting position (**B**,**C**). The right thoraco-lumbar inflexion is significantly reduced by the surgical fixation, but the rotation of the vertebral bodies is not totally corrected (**D**).

**Figure 2 biomedicines-12-00322-f002:**
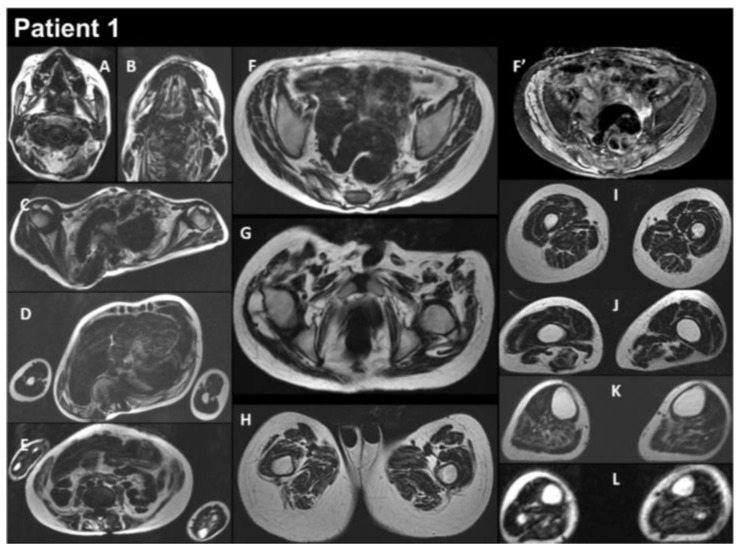
Selection of axial views performed from head to toes in a single multistacks MRI examination with a T2 Dixon sequence. All axial fat images (**A**–**L**) are comparable to T1-weighted images, and the water image (**F’**) is comparable to the T2-fat-saturated or STIR image. The examination showed that diffuse muscle atrophy was more pronounced in the paraspinal muscles, as well as in the abdominal belt and pelvic girdle areas. In the T2-fat-saturated image (water image, **F’**).

**Figure 3 biomedicines-12-00322-f003:**
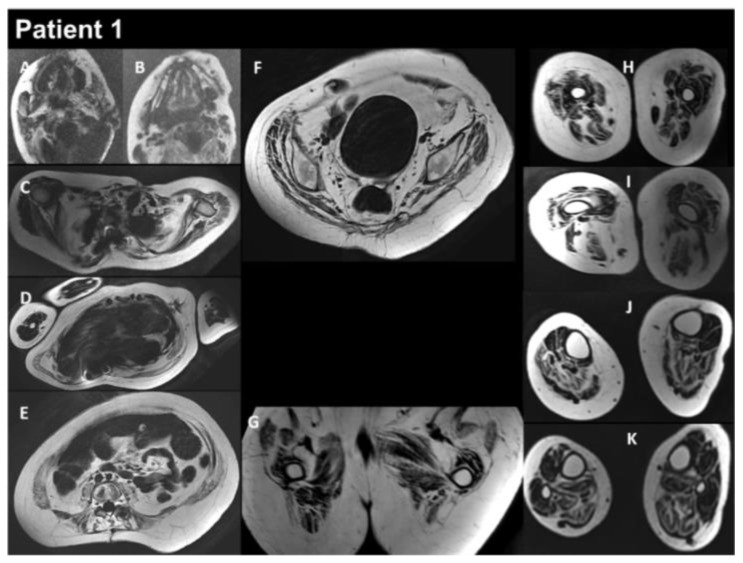
Selection of axial views performed from head to toes in a single multistacks MRI examination with a T2 Dixon sequence. All axial fat images (**A**–**K**) are comparable to T1-weighted images and performed at the same location as images A to L in Figure 2. The examination showed a severe increase in the atrophy of all muscles, as well as an increase in the fat contents of the muscles.

**Figure 4 biomedicines-12-00322-f004:**
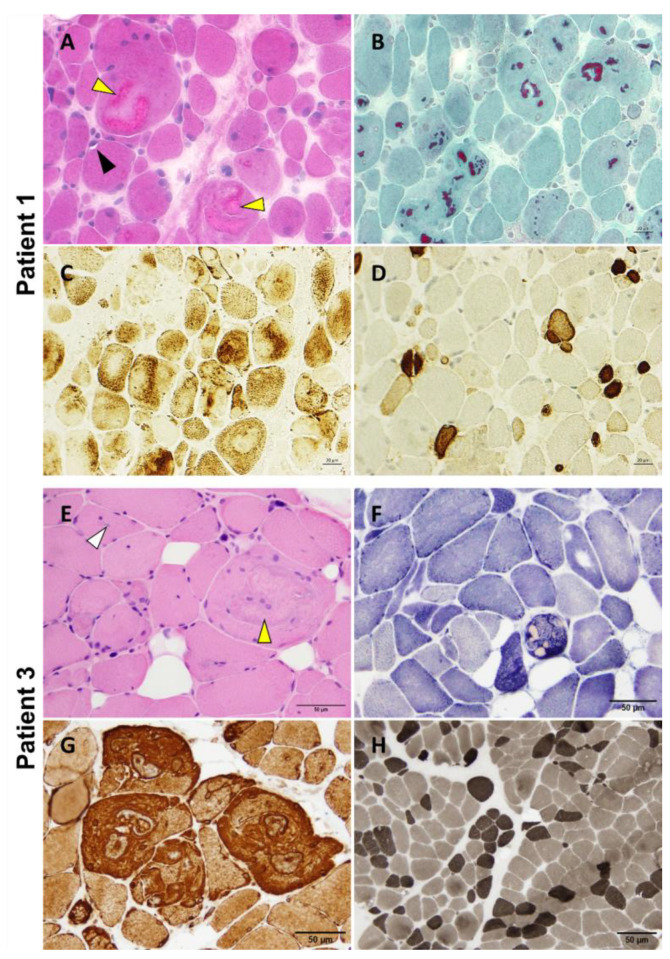
P1 and P3 light microscopy morphological analysis. Myopathic changes observed in these two patients include internalized nuclei, cytoplasmic bodies (yellow arrowhead), rimmed vacuoles (black arrowhead), and inclusion bodies (white arrowhead) (**A**,**E**). Cytoplasmic bodies appear to be fuchsinophilic with Gömöri trichrome staining (**B**) and are easily recognizable at oxidative staining as they misplace the mitochondrial network (**C**,**F**). Desmin reactivity was evident at the periphery of and in the areas surrounding the cytoplasmic bodies (**G**). Pronounced fiber size variability with type 2 fibers’ atrophy and type 1 fibers’ predominance are also observed (**D**,**H**). Samples were derived from muscles of P1 (**A**–**D**) and P3 (**E**–**H**). (**A**): H&E; (**B**): Gömöri trichrome; (**C**): COX; (**D**): Fast myosin immunohistochemistry; (**E**): H&E; (**F**): NADH; (**G**): Desmin staining; (**H**): ATPase 9.4. Scale bars: 20 µm (**A**–**D**) and 50 µm (**E**–**H**).

**Figure 5 biomedicines-12-00322-f005:**
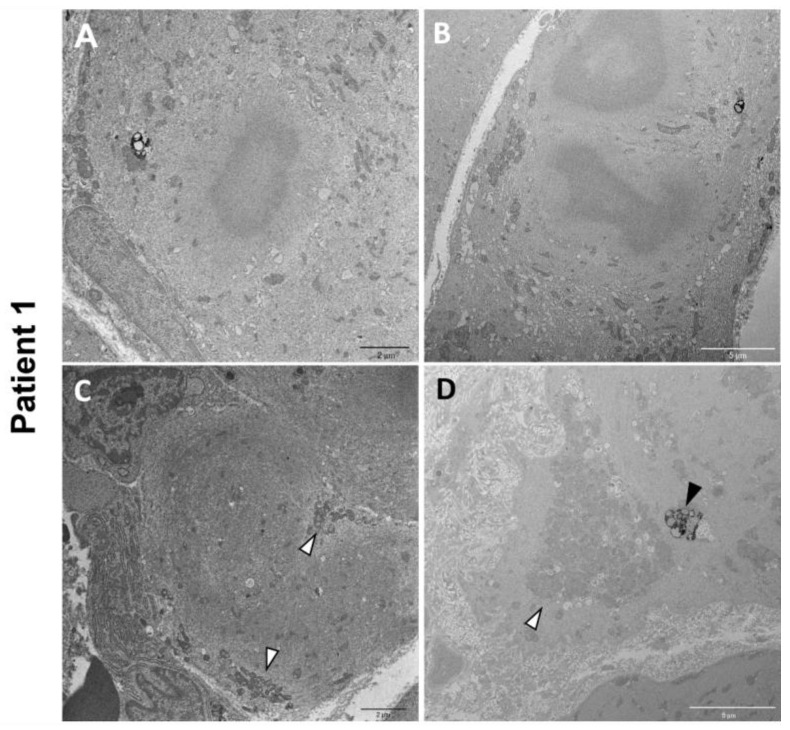
P1 electron microscopy ultrastructural analysis. Cytoplasmic bodies are visible in (**A**,**B**). Massive disorganization of the sarcomeric structure associated with clusters of mitochondria (white arrowheads) (**C**,**D**). Lipofuscin residual bodies are also observed (black arrowhead) (**D**).

**Figure 6 biomedicines-12-00322-f006:**
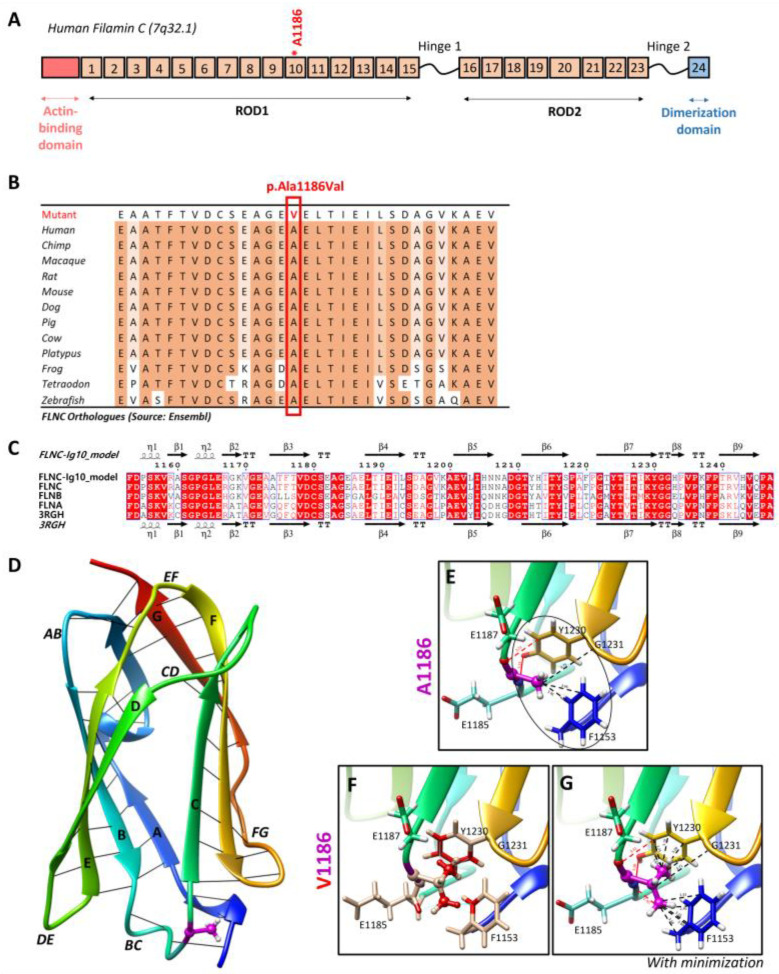
Two-dimensional and three-dimensional *FLNC* protein structures. (**A**) Schematic representation of human filamin C monomer with the actin-binding domain, different Ig domains forming Rods (ROD1, Ig1 to Ig15; ROD2, Ig16 to Ig23), and dimerization domain (Ig24). Hinges 1 and 2 are also indicated. Ig20 is bigger than other Ig domains because it presents an insertion of 83 amino acids compared to the other FLN proteins. The Ala1186 residue is located in Ig10 (shown by *). We note that hinge 1 of *FLNC* is spliced out during myogenesis. (**B**) The Ala1186Val missense variant is located in a conserved region among twelve different species. (**C**) Sequence alignment of the Ig10 domains of human FLNA, FLNB, and *FLNC*. The ESPript 3.0 server was used to output the alignment. The sequences’ and secondary structures’ depictions of our *FLNC*-Ig10 model and the FLNA-Ig10 (PDB: 3RGH) are also shown. The red box indicates a strict identity between all three FLN Ig10 protein sequences, while blue boxes indicate sequences presenting a similarity across at least two protein sequences. (**D**) Rainbow-colored 3D model of *FLNC*-Ig10 from N-terminus (blue) to the C-terminus (red). This model is based on the FLNA-Ig10 crystal structure (3RGH) using SWISS-MODEL. ꞵ-Strands are labeled from A to G, and interstrand loops are also labeled. The Ala1186 residue is shown in magenta, the side chain is represented as a ball and stick, and hydrogens atoms are represented as white balls. (**E**–**G**) Zoomed-in image of the Ala1186 residue and the mutated Val1186 residue, indicated in magenta. The residues at <5 Å of the 1186 residue are shown in stick representation with colored atoms (H in white, O in red, and N in blue) and are labeled. Hydrophobic interactions are shown by black lines and within the circle (dotted lines for carbon–carbon interactions and black circle for non-polar interactions), and hydrogen bonds are shown in red lines (dotted lines for weak C–H⋯O interactions and solid lines for strong N–H⋯O interaction). (**E**) Zoomed-in image of the Ala1186 residue and its interactions with nearby residues. (**F**) Zoomed-in image of the mutated Val1186 residue, and clashes with nearby residues are shown in red while contacts are shown as cyan lines. (**G**) Zoomed-in version of the mutated Val1186 residue after the minor minimization of mutated and nearby residues.

**Figure 7 biomedicines-12-00322-f007:**
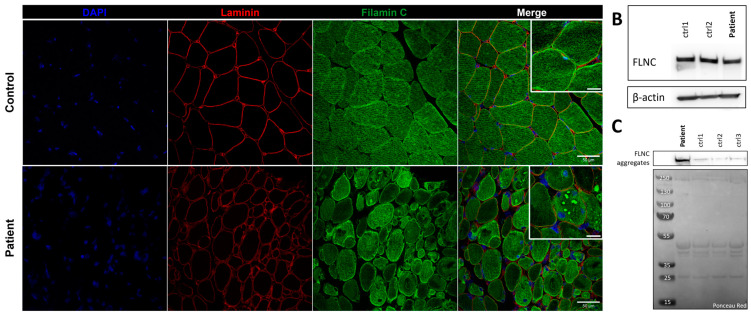
*FLNC* protein expression in muscle from P1 and control. (**A**) Immunofluorescence staining of nuclei (blue), laminin (red), and *FLNC* (green). Scale bar: 50 µm and 20 µm in zoomed-in images. (**B**) Soluble *FLNC* proteins in muscle homogenates from two controls and the patient. ꞵ-actin is used as the loading control. (**C**) Insoluble *FLNC* proteins representing *FLNC* aggregates in muscle biopsies from the patient and controls. Ponceau red is used as the loading control.

**Figure 8 biomedicines-12-00322-f008:**
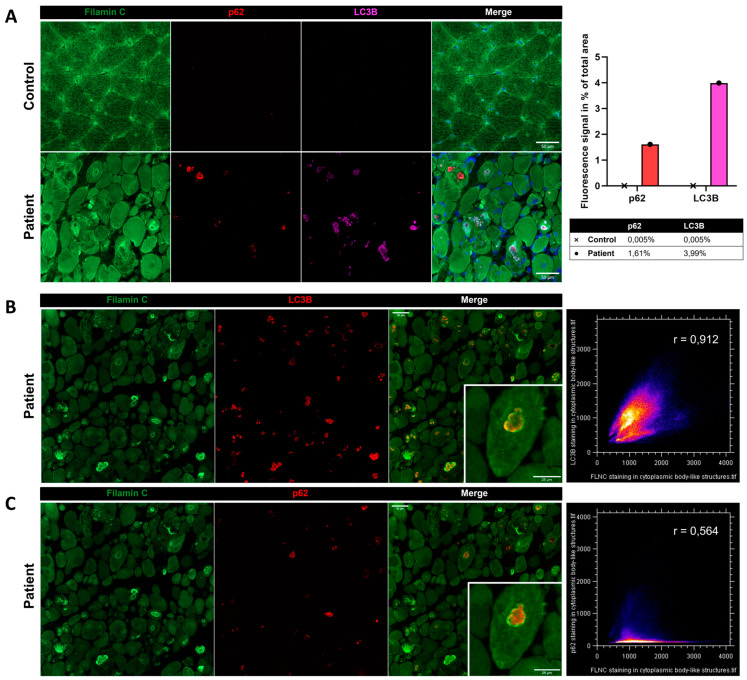
Immunofluorescence findings in muscle from P1 and the control. (**A**) Immunofluorescence analysis showing accumulations of *FLNC*, p62, and LC3B in myofibers from the patient’s muscle, which was not observed in the control. As shown by the histogram, we quantified the fluorescent signal of p62 and LC3B staining as the percentage of total area in the control’s and patient’s muscle biopsies. In the patient, LC3B colocalized with *FLNC* aggregates (**B**), whereas P62 accumulated inside the structure where *FLNC* accumulated but did not colocalize with *FLNC* (**C**). Colocalization analyses are presented as scatter plots with the Pearson coefficient (**B**,**C**). Scale bars: 50 µm and 25 µm in zoomed-in images (**B**,**C**).

**Table 1 biomedicines-12-00322-t001:** Comparison between previously reported patients and patients reported in this study with the *FLNC* Ala1186Val variant.

Ref	N°	Sexe	Status	Pathology	Onset		Neuromuscular Involvement				CK Levels	Cardiac Involv.	Resp. Involv.	Outcome	Anapath.
					Muscular	Cardiac	Muscle weakness	Arthrogryposis	Scoliosis	EMG					Presence of cytoplasmic bodies
[25]	P26	-	heteroz.	Myopathy	38 y		Proximal, mild distal	-	-	-	Increased	-	-	-	Yes
[10]	P25	M	heteroz., de novo	RCM + myopathy	Birth	1.4 y	Proximal	Yes	Yes	Myopathic	Increased CK 1.5–4	RCM	Respiratory infection at 2.5 y	Death at 2.5 y	-
	P16	F	heteroz.	RCM + myopathy	1st year	3 y	Proximal	-	-	Myopathic	Increased CK 1.2	RCM	-	Death at 9 y	-
	P22	F	heteroz., de novo	RCM + myopathy	Birth	15 y	Proximal	Yes	Yes	-	Normal	RCM	Pulmonary hypertension	Ht at 19 y	Yes
[26]	P1	M	heteroz., de novo	RCM	30 mo		-	-	-	-	Normal	RCM	Pulmonary hypertension	Last follow up at 40 mo	-
[23]	I298	M	heteroz., de novo	CM + myopathy	-		-	-	Yes	-	Normal	CM	Restrictive respiratory syndrome	Alive at 38 y	Yes
[24]	-	F	heteroz., de novo	HCM, MFM, multiorgan tumors	30 y	6 y	Proximal	-	Yes	Normal	Increased	HCM	FVC 24.5%	Alive at 41 y	Yes
[27]	D15-1576	F	heteroz., de novo	Arthrogryposis	-		-	Yes	Yes	-	-	-	-	-	-
[22]	P7	M	heteroz., de novo	RCM + mucoskeletal involvement	3.5y		-	Yes	Yes	-	Increased	RCM	-	Ht waiting list	-
	P8	M	heteroz., from father	RCM + mucoskeletal involvement	1y		-	-	Yes	-	Increased	RCM	-	Ht waiting list	-
[21]	P4	M	heteroz., de novo	RCM + myopathy	1st year	1 y	-	-	-	Sensory polyneuropathy in the legs and arms	Increased CK 1.32	RCM	No	-	-
	P8	M	heteroz., de novo	RCM + myopathy	Birth	3.5 y	Proximal	Yes	Yes	Normal	Increased CK 2.8	RCM	No	ICD	-
	P9	M	heteroz., de novo	RCM + myopathy	Birth	2 y	-	Yes	-	-	Increased CK 3.8	RCM	No	-	-
THIS STUDY	P1	M	heteroz., de novo	RCM + myopathy	Birth	13 y	Global	Yes	Yes	Myopathic	Normal	RCM	Restrictive respiratory syndrome	Alive at 17 y	Yes
	P2	M	heteroz.	HCM + myopathy	2 y	39 y	Global	-	Yes	-	Increased	HCM	Restrictive respiratory syndrome	Alive at 43 y	Yes
	P3	F	heteroz.	HCM + myopathy	3–4 y	29 y	Axial and proximal	-	Yes	-	Increased	HCM	Restrictive respiratory syndrome	Alive at 34 y	Yes

EMG, electromyography; CM, cardiomyopathy; RCM, restrictive cardiomyopathy; HCM, hypertrophic cardiomyopathy; MFM, myofibrillar myopathy; CK, creatine kinase; FVC, forced vital capacity; Ht, heart transplantation; ICD, implantable cardioverter defibrillator.

## Data Availability

The data that support the findings of this study are available on request from the corresponding author. The data are not publicly available due to privacy or ethical restrictions.

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
