# Peer review of "The FLNC Ala1186Val Variant Linked to Cytoplasmic Body Myopathy and Cardiomyopathy Causes Protein Instability"

_biomedicines, 2024, doi:10.3390/biomedicines12020322_

Round 1
Reviewer 1 Report
Comments and Suggestions for Authors
In the manuscript 'The FLNC p.Ala1186Val variant linked to cytoplasmic body myopathy and cardiomyopathy causes protein instability' submitted by Onnée et al. to Biomedicines, the authors identified a novel likely-pathogenic variant in FLNC and analyzed its impact.
The study is really interesting and the topic is highly relevant for a broad readership.
Nevertheless, there are some points which should be improved by the authors:
1.) I would use single or triple letter code for the amino acids instead of mixing them (see title and abstract).
2.) I suggest to write all human gene names in the complete manuscript in Italics.
3.) Not all abbreviations like PVDF are explained, when they were introduced. Please double check the complete manuscript.
4.) Abstract: Line 41: Please define the kind of cardiomyopathy here.
5.) Line 68. Please define HCM instead of general cardiomyopathies. Mutations in FLNC causing RCM were for example described for the first time in 2015 (Hum Mut).
6.) Chapter 2.6.2: Please indicate the used concentrations of the used primary and secondary antibodies. Please also indicate the DAPI concentration. Please indicate the specific secondary antibodies from Thermo Fisher Scientific. Which ones were used?
7.) Please use g x instead or rpm for centrifugation steps (e.g. line 180).
8.) Figure 4E-H: Scale bars are missing.
9.) Figure 5: Scale bars are missing.
10.) It is really interesting that the described mutation affects an amino acid localized at a loop between two beta sheets. The figure 7 explaining this is really convincing and helpful. Recently, two other RCM associated missense mutations (p.S1624L and p.I2160F) have been described, which are also localized at loop structures between beta-sheets. Therefore, I suggest to discuss this.
11.) I would also shortly discuss and compare FLNC with other genes, where mutations cause an aberrant cytoplasmic protein aggregation like DES (desmin) and CRYAB (alpha B Crystallin) leading to cardiomyopathies.
In summary, the authors presented a convincing study about a novel FLNC variant. However, the manuscript can be improved at different points. Therefore, I suggest a major revision. However, I am optimistic that the authors can update and improve their manuscript. Good luck!
Author Response
- I would use single or triple letter code for the amino acids instead of mixing them (see title and abstract).
- We thank the reviewer for this comment. We harmonized the amino acids mention by choosing the triple letter code.
- I suggest to write all human gene names in the complete manuscript in Italics.
- As suggested by the reviewer all the human gene names are in Italics
- Not all abbreviations like PVDF are explained, when they were introduced. Please double check the complete manuscript.
- We agree with the reviewer. We have verified and introduced all abbreviations mentioned all over the text.
- Abstract: Line 41: Please define the kind of cardiomyopathy here.
- We indicated the cardiomyopathy type which is “restrictive or hypertrophic cardiomyopathies” (Page 1, line 41 of revised manuscript)
- Line 68. Please define HCM instead of general cardiomyopathies. Mutations in FLNC causing RCM were for example described for the first time in 2015 (HumMut).
- We thank the reviewer for this comment. We changed the sentence accordingly for “Later, FLNC variants have also been associated with isolated cardiomyopathies [11-13]” and added the following references (doi:10.1161/CIRCGENETICS.113.000578 and doi:10.1002/humu.22942). (Page 2, lines 69-70 of revised manuscript) Of interest the 3 associated references (doi:10.1038/ncomms6326, doi:10.1161/CIRCGENETICS.113.000578 and doi:10.1002/humu.22942) are linked to the firsts FLNC-related isolated HCM, RCM or DCM.
- Chapter 2.6.2: Please indicate the used concentrations of the used primary and secondary antibodies. Please also indicate the DAPI concentration. Please indicate the specific secondary antibodies from Thermo Fisher Scientific. Which ones were used?
- We now inserted the concentration of FLNC, LC3B and p62 antibodies (page 4, line 152-153) and we inserted in the text that we used the following secondary antibodies: “anti-rabbit IgG-Alexa 488 (A11034), anti-mouse IgG1-Alexa 555 (A12227) and anti-mouse IgG2a-Alexa 647 (A21241), all from Thermo Fisher Scientific, 1:300” (Page 4, lines 154-156 of revised manuscript). We also added the DAPI concentration “1µg/mL” (Page 4, line 159 of revised manuscript). We apologize for this oversight.
- Please use g x instead or rpm for centrifugation steps (e.g. line 180).
- We changed the text accordingly at page4, lines 181 and 186 of revised manuscript.
- Figure 4E-H: Scale bars are missing.
- We apologize for this oversight; we added missing scale bars. (Page 8)
- Figure 5: Scale bars are missing.
- We now inserted the scale bars in figure 5. (Page 9)
- It is really interesting that the described mutation affects an amino acid localized at a loop between two beta sheets. The figure 7 explaining this is really convincing and helpful. Recently, two other RCM associated missense mutations (p.S1624L and p.I2160F) have been described, which are also localized at loopstructures between beta-sheets. Therefore, I suggest to discuss this.
- Thank you for this comment. We now included in the discussion the following sentence “Our results are consistent with another study that made a link between other variants located in the interstrands loops and cytoplasmic aggregates of FLNC in cardiac tissues in the context of RCM [13]. These findings highlight the importance of these loops for protein stability/folding whether in cardiac or skeletal muscle.” (Page 17, lines 507-511 of revised manuscript)
- I would also shortly discuss and compare FLNC with other genes, where mutations cause an aberrant cytoplasmic protein aggregation like DES (desmin)and CRYAB (alpha B Crystallin) leading to cardiomyopathies.
- We agree with the reviewer that this is an important point. As proposed, we discussed the similar protein aggregation observed in cardiomyopathies linked with DES and CRYAB mutations adding the following sentence in the text “We also evidenced in P3 an accumulation of desmin (DES) within or close to cytoplasmic bodies. Interestingly, mutations in DES often lead to the formation of intracellular aggregates, disrupting the normal architecture of muscle cells [13,37,38]. As discussed by Brodehl and colleagues, this aberrant aggregation is linked to a variety of myopathies and cardiomyopathies, similar to what is observed with FLNC mutations [13]. Alpha B Crystallin, encoded by the CRYAB gene, is a small heat shock protein that plays a crucial role in protein folding and aggregation inhibition. Mutations in CRYAB can disrupt its chaperone function, leading to protein misfolding and aggregation, which is a common pathologic feature in certain myopathies and cardiomyopathies [39,40]. Like FLNC and DES, the CRYAB mutations underscore the importance of protein stability and proper folding in maintaining cardiac muscle function. The comparison of FLNC with DES and CRYAB highlights a shared pathological mechanism in cardiomyopathies: the disruption of normal protein folding and aggregation processes. While each gene and its corresponding protein have unique roles and structures, the overarching theme of their contribution to cardiomyopathy lies in their failure to maintain cellular integrity due to aberrant protein aggregation. This comparison not only broadens our understanding of the molecular basis of cardiomyopathies but also suggests potential therapeutic targets centered around protein stability and aggregation control in these three conditions.” (Page 16, lines 444-462 of revised manuscript)
Reviewer 2 Report
Comments and Suggestions for Authors
This study evaluates the pathological impact of the FLNC c.3557C>T (p.Ala1186Val) variant, which is known to be associated with early-onset cytoplasmic body myopathy and cardiomyopathy. The research focuses on three unrelated patients showing a uniform clinical phenotype characterized by severe contractural myopathy leading to loss of gait, significant respiratory involvement, and cardiomyopathy. The study encompasses clinical imaging, myopathologic and genetic characterization, bioinformatics analysis, protein structure predication, and functional analysis. The FLNC A1186V variant is identified in the interstrands loop of filamin C, impacting its stability and leading to protein aggregation in the form of cytoplasmic bodies​​. Importantly, this study elucidates the impact of the p.Ala1186Val variant on FLNC protein structure, revealing alterations in beta sheet stability and interdomain interactions that likely contribute to protein aggregation in muscle fibers. Thus, this research provides a comprehensive understanding of the FLNC variant's impact on protein structure and function, underlining the need for further investigation into therapeutic approaches targeting these molecular mechanisms​​.
Here are the comments.
1. This study includes three unrelated patients with the FLNC c.3557C>T (p.Ala1186Val) variant. For figures 1 to 5, please label the corresponding patient number in the figure to make it clear to read.
2. For figures 4 and 5, please add the corresponding scale bar for the images. The EM images in Figure 5 look blur. Could the authors please provide more clearer EM images?
3. The authors can combine Figure 6 and Figure 7 into one figure. As the authors mentioned, the variant is a known pathogenic/likely pathogenic in ClinVar and has been reported multiple times in HGMD (PubMed: 36104822; PubMed: 35288587; PubMed: 35838873; PubMed: 36286284; PubMed: 34490615; PubMed: 34857437; PubMed: 34440373; …). Thus, there is no need for in silico prediction because the pathogenicity of this variant is known.
4. Could the authors please quantify the immunofluorescence findings in Figure 9?
5. Please improve the image quality of Figures 8 and 9, and make the scale bar clearer.
6. As mentioned above, the FLNC c.3557C>T (p.Ala1186Val) variant has been reported 12 times in HGMD database. There are some phenotypic differences, although the authors find their patients presented a homogeneous clinical phenotype. Could the authors please focus on the reported variant, summarize the phenotypes of the reported cases, and compare the previous findings with the current study?
Comments on the Quality of English LanguageMinor editing of English language required
Author Response
- This study includes three unrelated patients with the FLNC c.3557C>T (p.Ala1186Val) variant. For figures 1 to 5, please label the corresponding patient number in the figure to make it clear to read.
- Thank you for this comment, we added the patient number on the figures to improve the understanding of our results.
- For figures 4 and 5, please add the corresponding scale bar for the images. The EM images in Figure 5 look blur. Could the authors please provide more clearer EM images?
- We agree with the reviewers. Now all the images have the scale bars. Moreover, we selected new images for figure 5, with a better resolution; we did our very best to capture the most beautiful images possible. (Page 9)
- The authors can combine Figure 6 and Figure 7 into one figure. As the authors mentioned, the variant is a known pathogenic/likely pathogenic in ClinVar and has been reported multiple times in HGMD (PubMed: 36104822; PubMed: 35288587; PubMed: 35838873; PubMed: 36286284; PubMed:34490615; PubMed: 34857437; PubMed: 34440373; ...). Thus, there is no need for in silico prediction because the pathogenicity of this variant is known.
- As requested, we combined the figures 6 and 7. (Page 10)
- Thank you for this advice, we removed the in silico predictions for this variant. (Page 3, chapter 2.4 and page 9, chapter 3.3)
- Could the authors please quantify the immunofluorescence findings in Figure 9?
- We now quantified the immunofluorescent finding of p62 and LC3B expression in patient’s vs control’s muscle biopsies. We also added quantification of colocalization between FLNC and LC3B staining and FLNC and p62 staining using the Pearson’s coefficient. (Page 12)
- Please improve the image quality of Figures 8 and 9, and make the scale bar clearer.
- We enlarged the scale bars and improved images quality. (Page 12)
- As mentioned above, the FLNC c.3557C>T (p.Ala1186Val) variant has been reported 12 times in HGMD database. There are some phenotypic differences, although the authors find their patients presented a homogeneous clinical phenotype. Could the authors please focus on the reported variant, summarize the phenotypes of the reported cases, and compare the previous findings with the current study?
- We now added a table, table 1 (Pages 14-15), that report clearly the particularities of the phenotype of our 3 patients compared to the previously reported patients with the FLNC c.3557C>T (p.Ala1186Val). We added the following paragraph “The trajectory described in our three patients is consistent with the other previously re-ported cases whose characteristics are resumed in Table 1. In particular our cases show different age of onset with later onset cardiomyopathies, for both restrictive and hypertrophic.” (Page 13, lines 405-408)
Round 2
Reviewer 1 Report
Comments and Suggestions for Authors
The authors have improved their manuscript according to my suggestions. Congratulations! I suggest to accept this nice manuscript for publication.
Reviewer 2 Report
Comments and Suggestions for Authors
The manuscript has been significantly improved by the authors, and it looks good to me for the present version.